# Mortality after Sustaining Skeletal Fractures in Relation to Age

**DOI:** 10.3390/jcm11092313

**Published:** 2022-04-21

**Authors:** Camilla Bergh, Michael Möller, Jan Ekelund, Helena Brisby

**Affiliations:** 1Department of Orthopaedics, Institute of Clinical Sciences, Sahlgrenska Academy, University of Gothenburg, 413 45 Gothenburg, Sweden; michael.moller@vgregion.se (M.M.); helena.brisby@vgregion.se (H.B.); 2Department of Orthopaedics, Sahlgrenska University Hospital, 413 45 Gothenburg, Sweden; 3Centre of Registers Västra Götaland, 413 45 Gothenburg, Sweden; jan.ekelund@vgregion.se

**Keywords:** fracture register, fracture, mortality, osteoporosis, fragility fracture, AO/OTA classification

## Abstract

Age-related mortality across fractures in different anatomical regions are sparsely described, since most studies focus on specific age groups or fracture locations. The aim here was to investigate mortality at 30 days and 1 year post-fracture within four different age groups. All patients ≥ 16 years registered in the Swedish Fracture Register (SFR) 2012–2018 were included (*n* = 262,598 patients) and divided into four age groups: 16–49, 50–64, 65–79, and ≥80 years of age. Standardized mortality ratios (SMR) at 30 days and 1 year after sustaining a fracture were calculated using age- and gender-specific life tables from Statistics Sweden for each of the 27 fracture locations in the four age groups. Absolute mortality rates for the youngest age group for all locations were below 1% and 2% at 30 days and 1 year, respectively. For the patients in the two oldest age groups (65 and older), mortality rates were as high as 5% at 30 days and up to 25% at 1 year for certain fracture locations. For younger patients a few localizations were associated with high SMRs, whereas for the oldest age group 22 out of 27 fracture locations had an SMR of ≥2 at 30 days. Fractures of the femur (proximal, diaphysis, and distal) and humerus diaphysis fractures were among the fractures associated with the highest mortality rates and SMRs within each age group. Moderately high SMRs were further seen for pelvic, acetabulum, spine, and tibia fractures within all age groups. Regardless of age, any type of femur fractures and humerus diaphysis fractures were associated with increased mortality. In the oldest age groups, about twice as many patients died within 1 year after sustaining a fracture in almost any location, as compared with the expected mortality rates, whereas in the youngest age group only fractures in a few locations were associated with a high SMR.

## 1. Introduction

Comorbidities, bone quality, trauma type, and age may all influence the mortality risk after sustaining a fracture [1,2,3]. Proximal femur and humerus fractures, both common in elderly people [4,5,6,7], have been demonstrated to be associated with a relatively high mortality risk [8,9,10]. Other fractures of both the upper and the lower extremities are sparsely studied regarding mortality and often only reported for a specific age group or in relation to certain treatments, e.g., inpatient or outpatient hospital care [10,11]. 

Increased awareness of how different trauma types and locations of fractures in patients of different ages may impact mortality risk can provide useful information for planning and decision-making regarding treatment strategies, including risk-minimizing of surgical procedures, waiting time for surgery, and overall care. 

Mortality after an injury/disorder can be reported as the absolute risk of death in percentage during a certain time period or as a risk increase, usually reported as standardized mortality ratio (SMR). When comparing mortality risks in patients of different age groups, the rationale for focusing on SMRs is that there can be large differences in mortality during a certain time period (e.g., 1 year after injury) depending on age. However, it is, of course, also of interest to be aware of the absolute mortality rates in relation to an injury.

The main aim of the present study was to assess the 30-days and 1-year SMRs as well as absolute mortality rates following fractures in different locations for different age groups using data from a national quality register (the Swedish Fracture Register) with linkage to the Swedish Tax Agency population register. The second aim was to investigate the influence of age, gender, and fracture locations (divided between fractures considered as osteoporosis/osteopenia-related fractures or not) and trauma type (high/low energy) on mortality in fracture patients. 

## 2. Materials and Methods

### 2.1. Data Collection in the Swedish Fracture Register

The Swedish Fracture Register (SFR) started to collect data in 2011, and data collection has gradually expanded since then [12]. More than 90% of all departments treating fractures in Sweden are linked to the SFR [13], and 100% coverage is estimated for 2021. All fractures, both surgically and non-surgically treated, are prospectively registered, and the classification is performed using the Müller AO/OTA classification system [14,15] for most fracture locations. Independent validation studies regarding the fracture classification in the SFR have been performed [16,17,18]. When data from the SFR have been compared with the National Patient Register (NPR) at the Swedish Board of Health and Welfare, completeness of fracture registration has been found to be 70–95% for most participating departments [12]. Inclusion of patients’ Swedish personal identification number, consisting of an eight-digit date-of-birth number followed by a unique four-digit code, allows monitoring of patients over time and enables accurate linkage between national databases. Data on mortality for patients registered in the SFR are obtained by linkage to the Swedish Tax Agency population register. 

### 2.2. Patient Data

All patients 16 years and older who sustained a fracture that was registered in the SFR between the 1st of January 2012 and the 31st of December 2018 were included in the study. Only primary fractures were registered; re-fractures (a fracture within the same location during the study period) were excluded. One patient could have more than one fracture (different anatomical locations) registered during the study period. Fractures were divided into 27 anatomical regions according to the AO/OTA classification [14]. Registration of trauma mechanism (low, high, unknown, or undetermined) was performed at the primary patient registration. Classification of trauma mechanism was based on advice in the SFR user manual with reference to national trauma alert criteria; no systematic algorithm was used.

The patients were divided into four different age groups: 16–49, 50–64, 65–79, and ≥80 years. SMRs were calculated based on mortality among patients registered in the SFR and using the corresponding life tables for 2012–2018 retrieved from Statistics Sweden [19]. The life tables used reported the 1-year mortality rates for each year of age and for gender separately. When calculating the SMR for the 30-day period, this was performed under the assumption that the expected mortality for 30 days would be 1/12th of what was reported in the 1-year life tables. The dataset used in the present study has previously been used in a study on mortality rates in relation to different locations but with no analyses of different age groups [20].

### 2.3. Statistical Analyses

The SMR was calculated as the ratio between observed and expected mortality with 95% confidence intervals (CI), according to the method by Vandenbroucke [21]. All calculations were performed using SAS (v9.4, SAS Institute Inc., Cary, NC, USA)

### 2.4. Ethical Statement

The study was conducted in accordance with the ethical standard in the 1964 Declaration of Helsinki. The study was approved by the Central Ethical Review Board, Gothenburg (ID 792-17). All patients who are registered in the SFR receive information about their registration and are given the option of withdrawing.

## 3. Results

### 3.1. Baseline Data and Mortality Rate for All Fracture Types Per Age Groups

A total of 295,713 fractures were registered in the SFR during the study period. Within the age group of 16–49 years, a male dominance in fracture numbers was observed; however, in patients > 50 years old a majority of fractures, up to 74% in the oldest group (see Table 1), were observed in women. Irrespective of age, the overall 30-days and 1-year mortality rates for patients sustaining a fracture were higher compared with what would be expected in the general population (Table 1). The overall mortality rate within 30 days from fracture was less than 1% in patients < 65 years but close to 7% in patients ≥ 80 years. Regarding 1-year mortality rates, approximately 1% of the fracture patients with an age < 65 years died within one year compared with about 6% of the patients > 65 years and almost 25% for the patients ≥ 80 years of age (for details see Table 1).

For all age groups, the overall SMR at 30 days post-fracture was between 5–8. The overall SMR at 1 year post-fracture varied between 2 and 4, with the highest value observed in the youngest age group.

### 3.2. Mortality Rate and SMR for Different Fracture Locations in Patients 16 to 49 Years Old

The absolute mortality rate was low for all fracture locations in the youngest age group, with a 30-day mortality rate of ≤1% and a 1-year mortality rate of ≤2%.

The highest SMR at 30 days within the youngest age group was observed for injuries of the femur with proximal, diaphysis, and distal femur fractures, demonstrating an SMR of between 56 and 147 (mortality rate 0.5% and 0.8%, respectively). The majority of the femur fractures in this age group occurred in men, and diaphysis and distal femur fractures were mostly caused by high-energy trauma. For the proximal femur fractures, about one in four were associated with high-energy trauma (Figure 1a, Table 2. Other fractures with high SMRs at 30 days post-injury in this age group were pelvic fractures (SMR 90, mortality rate 0.5%), where a majority occurred in women and 50% were caused by high-energy trauma, as well as humerus diaphysis fractures (SMR 42, mortality rate 0.3%), occurring mostly in men and with 27% caused by high-energy trauma (Figure 1a and Figure 2a; Table 3 and Table 4).

The four fracture locations associated with the highest SMRs at 30 days post-injury also demonstrated the highest SMRs at 1 year after injury (SMR 14–26) in this age group. However, less marked differences were observed in the 1-year SMRs between these four fracture types and the six following fracture types (spine, pelvis, acetabulum, proximal tibia, diaphysis tibia, and distal humerus; SMR 9–13) than in the 30-day post-injury SMRs (Figure 1a and Figure 2a, Table 2, Table 3 and Table 4).

### 3.3. Mortality Rate and SMR for Different Fracture Locations in Patients 50 to 64 Years Old

The absolute mortality rate in this group was, at the most, 2% at 30 days (femur and humerus diaphysis fractures) and 8% (femur diaphysis fractures) at 1 year post-injury (Table 2 and Table 4).

The highest SMRs in the patients aged 50–64 years old were observed following fractures in different locations of the femur and of the humerus diaphysis, at both investigated time points, with an SMR between 27 and 48 at 30 days and between 10 and 16 at 1 year post-fracture. Fractures of the spine, acetabulum, pelvis, and proximal humerus were also associated with high SMRs (between 6 and 12 at 30-days and 5–7 at 1-year post-injury; see Figure 1b and Figure 2b and Table 2, Table 3 and Table 4).

In this age group, a large variation in gender distribution for different fracture locations was observed with a slight overall dominance of women (63%). The proportion of high-energy trauma was overall less than 10%, but for some fracture locations, a relatively large proportion of the fractures were caused by high-energy trauma, e.g., 23% in femur diaphysis fractures. Other fractures in this age group with relatively high proportions caused by high-energy trauma and showing high SMRs included spine, acetabulum, and pelvic fractures (Table 2, Table 3 and Table 4).

### 3.4. Mortality Rates and SMR for Different Fracture Locations in Patients 65 to 79 Years Old

The absolute mortality in this age group was ≤5% at 30 days for all fracture locations of the lower extremities, spine, and pelvis and <2.5% for upper extremity fractures. At 1 year post-injury, the highest absolute mortality rate was observed for femur diaphysis (16%) followed by proximal femur (15%) and distal femur (12%), as well as humeral diaphysis fractures (12%) (Table 2, Table 3 and Table 4).

The highest SMRs in the age group 65–79 years old were observed for fractures of the femur diaphysis, proximal and distal femur, and humerus diaphysis, with SMRs between 13 and 26 at 30 days and between 6 and 7 at 1 year after injury. Thereafter followed a number of fracture locations with relatively similar SMRs, both at 30 days (SMR 6–10) and 1 year (SMR 3–5) post-injury, including fractures of the spine, pelvis, acetabulum, tibia diaphysis, distal humerus, and clavicle. The most common fractures within this age group, distal radius fractures, demonstrated low SMRs (at 30 days 1.4 and at 1 year 1.0). See Figure 1c and Figure 2c and Table 2, Table 3 and Table 4.

In 22 of the 27 fracture locations, the majority of injuries in this age group occurred in women. For 8 of the locations, high-energy trauma was recorded in 10% or more of the injuries. In 4 of these 8 locations (with >10% high-energy trauma-associated injuries) a higher proportion of fractures were recorded for men, and in the remaining locations the gender proportions were close to equal (Table 2, Table 3 and Table 4).

### 3.5. Mortality Rate and SMR for Different Fracture Locations in Patients 80 Years Old or Above

The absolute mortality rates at 30 days post-injury for this age group were between 5–10% for spine, pelvic, and upper and lower extremity fractures, except for the most distal extremity fractures. At 1 year after fracture the absolute mortality rate, as expected, was high in this age group, between 10–30%. In this age group there was a dominance of women for all fracture locations. (Table 2, Table 3 and Table 4).

In patients > 80 years of age, the highest SMRs at 30 days post-fracture were observed for fractures of the proximal femur, femur diaphysis, and humerus diaphysis, all with SMRs around 9. Out of the 27 fracture locations, 13 were associated with an SMR between 4 and 10 at 30 days post-injury. (Figure 1d; Table 2, Table 3 and Table 4).

For 10 of the 27 fracture locations, SMRs of ≥2 (highest SMR 2.6) were observed at 1 year post-injury and included almost all fractures of the axial skeletal and of the proximal upper and lower extremities (to around the elbow and the ankle level). See Figure 2d and Table 2, Table 3 and Table 4.

In most locations, high-energy trauma cause was registered for a low proportion of the fractures (0–6%) (Table 2, Table 3 and Table 4).

## 4. Discussion

To our knowledge, this is the first study assessing mortality rates and standardized mortality ratios for different age groups of different fracture locations in the same well-defined adult population. The findings in the present study demonstrate that SMRs as well as absolute mortality rates vary substantially in relation to age and for different fracture locations. Fracture locations associated with the highest SMR within each age group are, however, consistent across age groups.

The overall SMR after sustaining any type of fracture was not markedly different for the different age groups, neither at 30 days nor at 1 year post-fracture. However, when comparing SMRs for different fracture locations, both similarities and differences were observed between the age groups. The highest SMR was observed in the youngest age groups depending, of course, on the extremely low risk for death in this group within a 1-year time interval. It is important to keep in mind that SMRs for different anatomical locations should only be compared within each age group and not between age groups. As expected, the absolute mortality rates demonstrated a reversed pattern, with highest absolute mortality rates observed after a fracture in the older age groups.

The femur diaphysis fractures demonstrated the highest or second highest SMRs of all locations within three of the four age groups, with exception of the oldest age group, at both studied time points. The percentage of patients who died within one year after sustaining a femur diaphysis fracture was, of course, much higher for the older patients (around 30%) compared with patients in the younger age groups (1–2%).However, the SMR was 147 for the youngest patients and 9 for patients > 80 years old. Increased mortality for femur diaphysis fracture patients has previously been reported in several studies with different age selections [8,11,22]. In an age group of 65 years and younger, Sommersalo et al. [11] showed that femoral diaphysis fractures had the highest mortality rate among fracture locations of the lower extremities, which is in agreement with findings from the present study. In addition, the other fracture locations of the femur (proximal and distal fractures) were among the fracture locations with the highest SMRs within all age groups at both studied time points. Regardless of underlying reasons, which may vary within different age groups, the findings in the present study stress the importance of the increase in mortality risk after sustaining any type of femur fracture, independent of patient age.

In addition to femur fractures, other fractures with high SMRs in the lower extremities or in close proximity, were tibia diaphysis, acetabular, and pelvic fractures. For tibia diaphysis fractures, the risk increase was most pronounced in the older ages compared with other fractures within each age group even if the SMRs were highest in the youngest age group. For the oldest age group, the mortality risk following tibia diaphysis fractures was similar to that of proximal femur fractures. For pelvic and acetabular fractures, the SMRs were relatively high for all age groups, however the acetabular fractures were relatively few and thus with less reliable results. 

Overall, in all age groups only a few of the fracture locations in the upper extremities demonstrated high SMRs and low absolute mortality rates compared with fractures of the axial skeletal and the lower extremities. However, humerus diaphysis fractures were among the five fracture locations with the highest SMRs at both time points in all age groups and, interestingly, were comparable to proximal femur fractures regarding SMRs within all age groups, at both time points. There has recently been a detailed report published on the SMR of proximal humerus fractures, based on data from the same register as the present study [23]. In previous studies the absolute mortality rate for proximal humerus has been in accordance with the findings in the present study, with about 10% in patients > 65 years at 1 year post-fracture [24,25]. 

An observation when comparing age groups was that differences in SMRs for different fracture locations within each age group were less pronounced in the older age groups, e.g., the fracture location seemed to play less of a role in relation to mortality risk than in younger patients. In the oldest patient group, fractures in any location, except for the most peripheral parts of the extremities, were associated with an SMR of similar magnitude (SMR around 2 at 1 year after fracture) indicating that most fractures in this age group may be considered as frailty markers [3,26]. It is worth noting that the much lower number of fractures, as well as the low mortality rates, in the youngest patients makes the results for this group less reliable (large confidence intervals) than in the older age groups and need to be interpreted with some caution.

The strength of the present study was that the mortality rate and SMR could be described for the different fracture locations in different age groups for an adult population. A limitation was, despite the overall large number of included fractures, the relatively low number of fractures in certain locations and age groups, resulting in less reliable figures for some combinations of age and fracture locations. Another limitation was that each fracture was studied in relation to mortality independently if the patient suffered from multiple fractures; however, in 97% of the registered injury occasions only one fracture was registered.

In summary, all types of femur fractures resulted in high SMRs within each of the four age groups. Other fractures with high SMRs within each age group at both investigated time points were fractures of humerus diaphysis, acetabulum, pelvis, and spine. Bearing this in mind, patients with these fractures should be treated with caution. In addition, it can be concluded that, regardless of age, any type of femur fracture and humerus diaphysis fracture is associated with the highest absolute mortality figures within the respective age groups. It is important to remember that, in the oldest patients, sustaining almost any fracture results in twice as many deaths as compared with what is expected within a year. Awareness of both SMRs and absolute mortality rates for different fracture locations within different age groups are of importance in the prioritizing and organization of care for these patients.

## Figures and Tables

**Figure 1 jcm-11-02313-f001:**
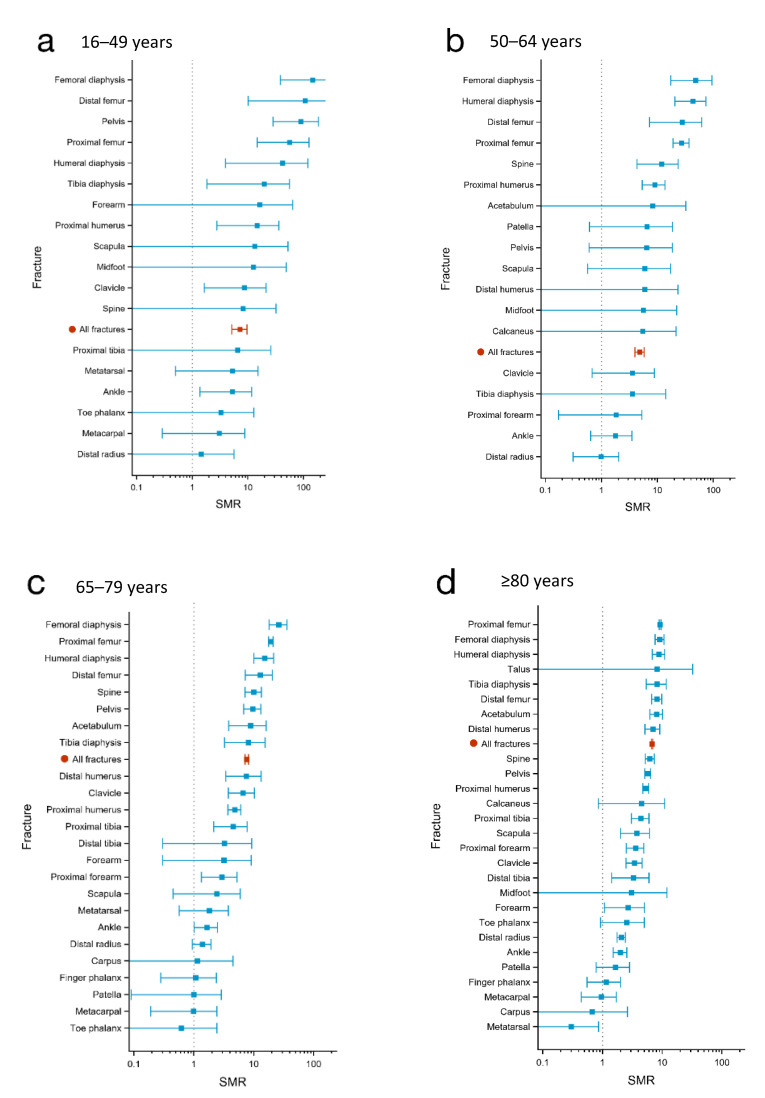
30-day standardized mortality ratios (SMR) for the different fracture locations in four age groups; (**a**): 16–49 years; (**b**): 50–64 years; (**c**): 65–79 years; and (**d**): ≥80 years. The dots represent SMR, and the horizontal lines represent the 95% confidence intervals.

**Figure 2 jcm-11-02313-f002:**
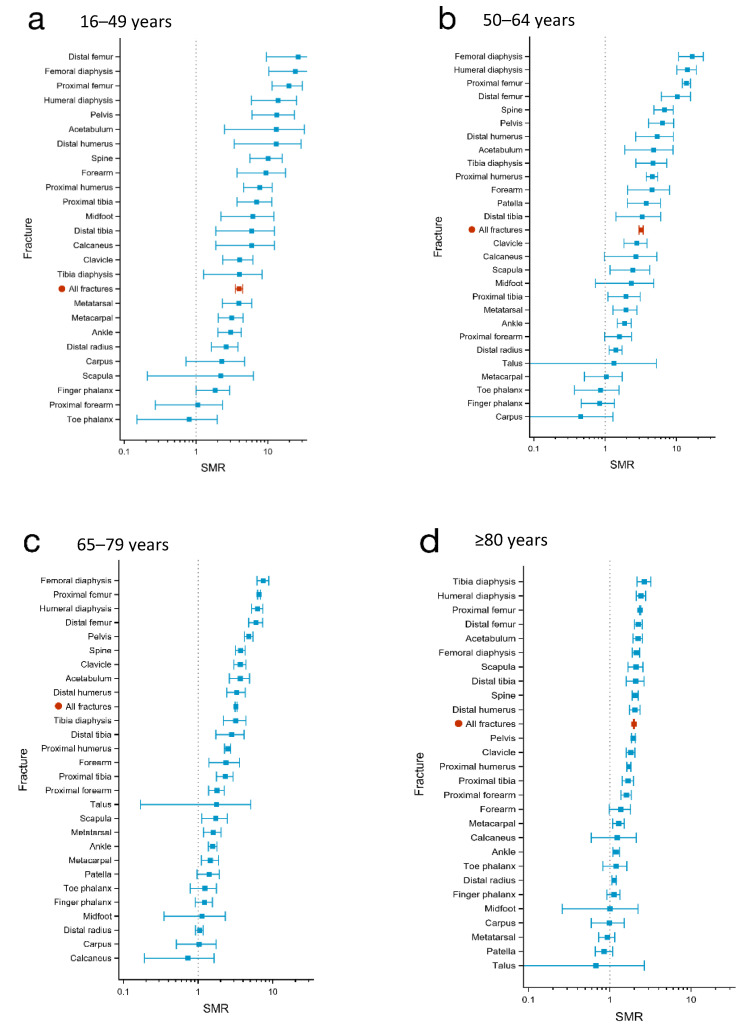
One-year standardized mortality ratios (SMR) in the different fracture locations in four age groups; (**a**): 16–49 years; (**b**): 50–64 years; (**c**): 65–79 years; and (**d**): ≥80 years. The points represent SMR and the horizontal lines represent the 95% confidence intervals.

**Table 1 jcm-11-02313-t001:** Baseline data, proportion deceased within 30 days and 1 year after sustaining a fracture, and Standardized Mortality Ratio (SMR) with 95% CI, for different age groups.

Fracture Location	Age Years	Number of Fractures	Women %	High Energy %	Dead within 30 Days %	Dead Expected 30 Days %	SMR 30 Days (95% CI)	Dead within 1 Year %	Dead Expected 1 Year %	SMR 1 Year (95% CI)
All fractures	16–49	89,075	36.0	16.6	0.04	0.006	7.2 (5.2–9.6)	0.3	0.1	4.0 (3.5–4.5)
	50–64	61,825	62.8	9.2	0.2	0.004	4.8 (4.0–5.8)	1.4	0.4	3.2 (3.0–3.4)
	65–79	74,297	69.6	4.3	1.2	0.2	7.6 (7.2–8.2)	6.1	1.9	3.2 (3.1–3.3)
	≥80	70,516	74.4	1.0	6.8	1.0	6.8 (6.6–7.0)	24.5	12.4	2.0 (2.0–2.0)

**Table 2 jcm-11-02313-t002:** Number of fractures, proportion of women (% total) and high-energy trauma (% total), mortality rate per time point after fracture, and standardized mortality rate (SMR) per time point after fracture for lower extremity injuries by age group.

Age Years		Aceta-Bulum	Femur Prox.	Femur Diaph.	Femur Distal	Patella	Tibia Prox.	Tibia Diaph.	Tibia Distal	Ankle	Talus	Calca-Neus	Mid-Foot	Meta-Tarsale	Toe Phalanx
**16–49**	Number of fractures	251	810	474	291	871	2644	1596	1092	11,912	683	1022	1363	6381	5099
	Women %	23	31	24	35	36	40	30	36	46	36	28	36	44	46
	High energy %	79	27	70	34	17	28	32	32	9	42	40	24	7	8
	Dead at 30 days %	0.0	0.5	0.8	0.7	0.0	0.0	0.1	0.0	0.0	0.0	0.0	0.1	0.0	0.0
	Dead at 1-year %	1.2	2.1	1.7	2.1	0.0	0.6	0.3	0.5	0.2	0.0	0.5	0.4	0.3	0.1
	SMR 30 days	0.0	56.4	146.9	107.1	0.0	6.6	19.6	0.0	5.3	0.0	0.0	12.5	5.3	3.3
	SMR 1 year	13.1	19.7	24.1	26.4	0.0	7.0	4.0	6.0	3.0	0.0	6.0	6.2	3.9	0.8
**50–64**	Number of fractures	277	3009	302	359	813	1829	746	529	9574	187	511	525	3303	2308
	Women %	28	52	50	60	66	63	43	48	63	46	40	52	66	62
	High energy %	45	7	23	11	5	17	21	24	4	33	31	19	5	6
	Dead at 30 days %	0.4	1.2	2.0	1.1	0.2	0.0	0.1	0.0	0.1	0.0	0.2	0.2	0.0	0.0
	Dead at 1-year %	2.5	7.4	8.3	5.0	1.7	0.8	2.1	1.5	0.8	0.5	1.2	1.0	0.8	0.3
	SMR 30 days	8.2	27.1	48.2	27.9	6.5	0.0	3.6	0.0	1.8	0.0	5.5	5.6	0.0	0.0
	SMR 1 year	4.7	13.8	16.5	10.3	3.7	2.0	4.7	3.3	1.9	1.3	2.7	2.3	2.0	0.9
**65–79**	Number of fractures	477	14,779	726	720	1306	1566	582	444	8381	73	308	278	2023	1144
	Women %	30	62	64	74	74	74	53	59	68	56	46	52	73	60
	High energy %	19	1	4	5	2	9	13	14	3	27	29	14	3	4
	Dead at 30 days %	1.7	3.6	4.7	2.1	0.2	0.6	1.2	0.4	0.2	0.0	0.0	0.0	0.2	0.1
	Dead at 1-year %	8.4	14.8	16.1	11.7	2.6	4.0	5.7	4.7	2.7	2.7	1.3	1.8	2.6	2.1
	SMR 30 days	8.9	19.3	26.2	12.9	1.0	4.5	8.2	3.2	1.7	0.0	0.0	0.0	1.8	0.6
	SMR 1 year	3.6	6.5	7.4	5.9	1.4	2.3	3.2	2.8	1.6	1.8	0.7	1.1	1.6	1.2
**80+**	Number of fractures	713	32,757	1284	1106	710	789	309	218	3108	17	78	46	768	294
	Women %	53	70	79	88	71	80	78	81	76	77	68	70	84	58
	High energy %	2	<1	<1	<1	<1	4	6	4	2	29	18	2	2	2
	Dead at 30 days %	9.1	10.0	10.0	8.9	1.4	4.3	8.4	3.6	1.7	5.9	3.8	2.2	0.3	2.0
	Dead at 1-year %	30.4	31.2	28.1	29.8	8.9	20.0	33.0	28.0	13.1	5.9	12.8	8.7	9.8	11.6
	SMR 30 days	8.1	9.2	9.1	8.1	1.6	4.4	8.2	3.3	2.0	8.2	4.5	3.0	0.3	2.6
	SMR 1 year	2.2	2.4	2.1	2.2	0.8	1.7	2.6	2.1	1.2	0.7	1.2	1.0	0.9	1.2

**Table 3 jcm-11-02313-t003:** Numbers of fractures, proportion of women (% total) and high-energy trauma (% total), mortality rate per time point after fracture, and standardized mortality ratios (SMR) per time point after fracture for spine and pelvic injuries by age group.

Age Years			Spine	Pelvis
**16–49**	Number of fractures		1892	996
	Women %		34	61
	High energy %		55	48
	Dead at 30 days %		0.1	0.5
	Dead at 1-year %		0.8	0.9
	SMR 30 days		8.2	89.8
	SMR 1 year		10.1	13.3
**50–64**	Number of fractures		1306	840
	Women %		42	61
	High energy %		34	30
	Dead at 30 days %		0.5	0.2
	Dead at 1-year %		3.1	2.9
	SMR 30 days		12.0	6.4
	SMR 1 year		6.7	6.3
**65–79**	Number of fractures		2221	2176
	Women %		48	72
	High energy %		15	8
	Dead at 30 days %		1.8	1.7
	Dead at 1-year %		8.1	10.2
	SMR 30 days		10.0	9.7
	SMR 1 year		3.7	4.7
**80+**	Number of fractures		2239	4781
	Women %		62	82
	High energy %		4	1
	Dead at 30 days %		6.0	6.1
	Dead at 1-year %		24.0	25.4
	SMR 30 days		6.2	5.7
	SMR 1 year		2.0	1.9

**Table 4 jcm-11-02313-t004:** Number of fractures, proportion of women (% total) and high-energy trauma (% total), mortality rate per time point after fracture, and standardized mortality ratios (SMR) per time point after fracture for upper extremity injuries by age group.

Age Years		Scapula	Clavicle	Humerus Prox.	Humerus Diaph.	Humerus Distal	Forearm Prox.	Forearm	Radius Distal	Carpus	Meta-Carpale	Finger Phalanx
**16–49**	Number of fractures	942	5339	2644	762	479	5158	1087	11,327	2969	11,347	10,022
	Women %	16	19	43	32	54	44	26	50	18	16	32
	High energy %	39	29	16	27	21	11	30	14	14	9	13
	Dead at 30 days %	0.1	0.1	0.1	0.3	0.0	0.0	0.1	0.0	0.0	0.0	0.0
	Dead at 1-year %	0.2	0.3	0.7	1.0	0.8	0.1	0.6	0.2	0.2	0.2	0.1
	SMR 30 days	13.4	8.7	14.7	42.0	0.0	0.0	16.3	1.4	0.0	3.1	0.0
	SMR 1 year	2.2	4.0	7.7	13.8	13.0	1.0	9.4	2.6	0.4	3.1	1.8
**50–64**	Number of fractures	875	2283	5379	604	440	3327	442	14,685	1030	2482	3860
	Women %	36	35	73	58	63	71	46	80	50	48	44
	High energy %	22	23	5	10	10	6	26	4	7	10	13
	Dead at 30 days %	0.2	0.1	0.3	1.7	0.2	0.1	0.0	0.0	0.0	0.0	0.0
	Dead at 1-year %	1.1	1.2	2.1	6.6	2.5	0.6	2.0	0.6	0.2	0.4	0.4
	SMR 30 days	6.0	3.6	9.0	43.0	5.9	1.8	0.0	1.0	0.0	0.0	0.0
	SMR 1 year	2.4	2.8	4.6	14.1	5.4	1.6	4.5	1.4	0.5	1.0	0.8
**65–79**	Number of fractures	779	1553	9179	1123	755	2212	406	16,073	589	1987	2437
	Women %	48	50	79	67	73	75	60	84	58	63	48
	High energy %	14	13	2	3	5	4	13	2	5	5	12
	Dead at 30 days %	0.4	1.0	0.7	2.4	1.2	0.4	0.5	0.2	0.2	0.2	0.2
	Dead at 1-year %	3.3	7.0	4.5	11.9	6.4	3.0	4.4	1.8	1.9	2.7	2.3
	SMR 30 days	2.4	6.6	4.8	15.2	7.5	3.0	3.2	1.4	1.1	1.0	1.1
	SMR 1 year	1.7	3.6	2.5	6.2	3.3	1.8	2.4	1.0	1.0	1.5	1.2
**80+**	Number of fractures	361	1144	6370	778	705	1004	269	8525	190	1005	948
	Women %	66	66	82	73	77	77	78	86	58	67	62
	High energy %	6	3	<1	1	1	1	5	<1	<1	2	3
	Dead at 30 days %	3.6	3.5	4.8	8.4	6.7	3.4	2.6	1.8	0.5	0.9	1.1
	Dead at 1-year %	24.4	22.3	19.0	28.5	23.7	18.2	16.0	12.1	9.5	14.5	12.3
	SMR 30 days	3.8	3.4	5.3	8.8	7.0	3.6	2.7	2.1	0.7	1.0	1.2
	SMR 1 year	2.1	1.8	1.7	2.4	2.0	1.6	1.4	1.1	1.0	1.3	1.1

## Data Availability

Data supporting the results are presented within the study. Data was obtained from The Swedish Fracture Register, the Swedish Tax Agency population register and Statistics Sweden and licensed to the authors for this study, why restrictions apply to the availability of these data.

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
