# Peer review of "Mortality after Sustaining Skeletal Fractures in Relation to Age"

_jcm, 2022, doi:10.3390/jcm11092313_

Round 1
Reviewer 1 Report
General comments: This manuscript represents a retrospective analysis of standardized mortality ratios (SMR) and absolute mortality stratified across age groups, sexes, and skeletal sites based on data obtained from the Swedish Fracture Register database. The manuscript and corresponding data add novel insights to the literature and will be of interest to a general readership, but a few issues require correction and clarification as described below.
Suggestions for authors:
- Results, lines 120-122: The authors have inadvertently left a few sentences from the template document in their submitted manuscript, these should be removed and the manuscript carefully proofread throughout.
- Results, general: Each sub-section of the results is inexplicably labeled as “section 3.1”, this should be corrected to present in numerical order (i.e., 3.1 to 3.6)
- Table 1: The choice to list only 1 significant digit and then present values showing as “0.0” as a footnote is confusing. Please simply increase the number of significant digits shown for “Dead within 30 days %” and “Dead expected 30 days” to accommodate the number of significant digits needed for all rows and eliminate the footnote.
- Results, section 3.6 “Impact of different potential risk factors on mortality risk…” section and Table 5: The authors have changed their groupings in Table 5 compared to what is presented in Tables 1-4, in that the group “16-49” is presented in Table 5 as separate rows for “16-20”, “21-30”, etc. Please re-formulate Table 5 so that it is consistent with the earlier tables and experimental design for the study, or otherwise the connection between claims made in Table 5 to the rest of the study is difficult (if not impossible) to make.
Also, there are several typographical errors in Table 5. The OR for “non fragility fracture”, “Women”, “Low energy”, and “91-108” are listed as “Reference” rather than a number. This must be corrected.
Also, the reference in this text to the table shows some disagreement, for example when referring to risk of mortality at 1 year after fracture, the authors point out an odds increase of 19-fold for ages 51-60 which matches the table, but then describe 49 fold for the 61-70 group when only 0.046 is shown. Please address / correct as necessary.
Reviewer 2 Report
The aim was to investigate mortality at 30-days and 1-year post-fracture within four different age groups.
Page 1: edit page 1 according to MDPI template.
Line 67: add the citation of database reliability of The Swedish Fracture Register.
Line 83: If patient have multiple fractures, how the patient was divide into the category?
Line 102: If you assess the mortality in patients with fracture, add the CCI variables.
Line 103: describe osteoporosis and osteopenia. I think that there are measurement bias.
Line 110: describe the definition of Explanatory factor in logistic model. Which varibles? Why did you select these variables?
Line 124: describe the number of patients with follow up loss
Line 336: : please write clinical implication clearly.
Line 338: : please write your limitation clearly through your research. For example, selection bias, information bias, confounding factor. I think that there are many bias in your study.
Line 341: how your results change the clinical practice?
Author Response
Please see the attchment

Round 2
Reviewer 2 Report
1: Add the document "Answers to my question".
I cannot find it.
2: In revised manuscript, keep the track changed
Author Response
Thank you for your thorough review of our manuscript. See our answers and suggested changes based on your comments.
